# Analgesic Efficacy of a Combination of Fentanyl and a Japanese Herbal Medicine “*Yokukansan*” in Rats with Acute Inflammatory Pain

**DOI:** 10.3390/medicines7120075

**Published:** 2020-12-17

**Authors:** Yuko Akanuma, Mami Kato, Yasunori Takayama, Hideshi Ikemoto, Naoki Adachi, Yusuke Ohashi, Wakako Yogi, Takayuki Okumo, Mana Tsukada, Masataka Sunagawa

**Affiliations:** 1Department of Physiology, School of Medicine, Showa University, Tokyo 142-8555, Japan; yukoaka@med.showa-u.ac.jp (Y.A.); jt.kato0513@med.showa-u.ac.jp (M.K.); ytakayama@med.showa-u.ac.jp (Y.T.); h_ikemoto@med.showa-u.ac.jp (H.I.); nadachi@med.showa-u.ac.jp (N.A.); denta1986@gmail.com (Y.O.); wkkyg0613@cmed.showa-u.ac.jp (W.Y.); tokumo@med.showa-u.ac.jp (T.O.); m-tsukada@med.showa-u.ac.jp (M.T.); 2Department of Anesthesiology, St. Luke’s International Hospital, Tokyo 104-8560, Japan; 3Department of Palliative Medicine, Showa University Yokohama Northern Hospital, Kanagawa 224-8503, Japan; 4Pharmaceutical Department, Showa University Hospital, Tokyo 142-8666, Japan

**Keywords:** *Yokukansan*, fentanyl, transient receptor potential ankyrin 1 (TRPA1) channel, phosphorylated extracellular signal-regulated kinase (pERK), whole-cell patch-clamp recording, herbal medicine

## Abstract

**Background:** Fentanyl can induce acute opioid tolerance and postoperative hyperalgesia when administered at a single high dose; thus, this study examined the analgesic efficacy of a combination of fentanyl and *Yokukansan* (YKS). **Methods:** Rats were divided into control, formalin-injected (FOR), YKS-treated+FOR (YKS), fentanyl-treated+FOR (FEN), and YKS+FEN+FOR (YKS+FEN) groups. Acute pain was induced via subcutaneous injection of formalin into the paw. The time engaged in pain-related behavior was measured. **Results:** In the early (0–10 min) and intermediate (10–20 min) phases, pain-related behavior in the YKS+FEN group was significantly inhibited compared with the FOR group. In the late phase (20–60 min), pain-related behavior in the FEN group was the longest and significantly increased compared with the YKS group. We explored the influence on the extracellular signal-regulated kinase (ERK) pathway in the spinal cord, and YKS suppressed the phosphorylated ERK expression, which may be related to the analgesic effect of YKS in the late phase. **Conclusions:** These findings suggest that YKS could reduce the use of fentanyl and combined use of YKS and fentanyl is considered clinically useful.

## 1. Introduction

Fentanyl and remifentanil are potent ultra-short-acting μ-opioid receptor agonists widely used for pain management in the perioperative period [1]. However, their use is limited because they can induce acute opioid tolerance and a post-treatment state of heightened pain sensitivity, known as opioid-induced hyperalgesia (OIH), when administered at a single high dose [2,3,4,5]. Development of OIH causes several problems, including delayed recovery after surgery, higher consumption of analgesics, and side effects associated with their administration. Remifentanil-induced hyperalgesia (RIH) has been extensively investigated [6,7,8,9,10,11]. There is some evidence that glutamate release and N-methyl-D-aspartate (NMDA) receptor activation may be important in the development of RIH [9,10,11]. In addition, remifentanil infusion downregulates µ-opioid receptors [12]. Moreover, glial activation is involved in RIH. Microglia and astrocytes are activated by chronic opioid use, and their inhibition seems to reduce RIH [6,13]. Some clinical [14,15,16,17] and basic studies [18,19,20,21,22] reported that high-dose fentanyl use may also cause hyperalgesia. Xuerong et al. [14] conducted a randomized controlled trial on 90 women undergoing total abdominal hysterectomy. Patients in the control group who were administered bupivacaine requested significantly less morphine postoperatively compared with patients treated with fentanyl. However, the combined use of ketamine, an NMDA receptor antagonist, and fentanyl significantly decreased the need for postoperative morphine administration. Richebé et al. [18] reported a similar phenomenon in fentanyl-treated rats. These results suggest that NMDA receptor activation is involved in the development of fentanyl-induced hyperalgesia (FIH).

*Yokukansan* (YKS) is a Japanese traditional herbal (Kampo) medicine that comprises seven herbs (Table 1). YKS is officially approved as an ethical pharmaceutical by the Japanese Ministry of Health, Labor, and Welfare. The three-dimensional high-performance liquid chromatography (3D-HPLC) profile chart of YKS was provided by Tsumura & Co. (Figure 1). YKS is administered to patients with symptoms, such as emotional irritability, neurosis, and insomnia, and to infants who suffer from night crying and convulsions [23,24]. Recently, it has been reported that YKS is effective against pain disorders, including headache, post-herpetic neuralgia, fibromyalgia, phantom-limb pain, and trigeminal neuralgia [25,26,27,28]. Studies have demonstrated antinociceptive effects of YKS in animal models with chronic neuropathic and inflammatory pain [29,30,31,32]. We previously reported that pre-administration of YKS attenuated the development of antinociceptive morphine tolerance and that suppression of glial cell activation may be one mechanism underlying this phenomenon [33,34]. YKS is also known to have an ameliorative effect on glutamate clearance in astrocytes and an antagonistic action at the NMDA receptor [35,36,37]. As mentioned above, NMDA receptor activation may be involved in FIH development [14,18]. Thus, we hypothesized that YKS might inhibit FIH development.

In the present study, we first evaluated the effect of combined treatment with YKS and fentanyl using the well-established inflammatory pain model induced by formalin injection [38]. The injection of formalin into the plantar surface of rodent paws induces acute nociceptive responses, such as lifting, licking, and flinching of the paw, which are biphasic. In the initial period of about 10 min (phase I), behavioral responses occur due to activated primary afferent nerve terminals and are mediated by activation of the transient receptor potential ankyrin 1 (TRPA1) channel [39,40]. Thus, we performed whole-cell patch-clamp recording in HEK293T cells expressing human TRPA1 to assess the influence of YKS or fentanyl on the TRPA1 channel. Phase II (10–60 min) reflects central sensitization of neurons in the dorsal horn and peripheral sensitization of nociceptors by the formalin-induced inflammatory response [41]. Accordingly, the expression of phosphorylated extracellular signal-regulated kinase (pERK) in the spinal dorsal horn was analyzed by immunofluorescent staining. ERK is an important molecule in pain signaling and a potential novel target for pain treatment [42].

## 2. Materials and Methods

### 2.1. Animals

Experiments were performed using 7–8-week-old male Wistar rats (Nippon Bio-Supp. Center, Tokyo, Japan). Animals were housed two to three per cage (W 24 × L 40 × H 20 cm) under a 12 h light/dark cycle in our animal facility at 25 °C ± 2 °C and 55% ± 5% humidity. Food (CLEA Japan, CE-2, Tokyo, Japan) and water were provided ad libitum. The experiments were performed in accordance with the guidelines of the Committee of Animal Care and Welfare of Showa University. All experimental procedures were approved by the Committee of Animal Care and Welfare of Showa University (certificate number: 07064, date of approval: 1 April 2017). Effort was made to minimize the number of animals used and their suffering.

### 2.2. Administration of Drugs

Dry powdered extracts of YKS (Lot No. 2110054010) used in the present study were supplied by Tsumura & Co. (Tokyo, Japan). The seven herbs comprising YKS (Table 1) were mixed and extracted with purified water at 95.1 °C for 1 h; the soluble extract was separated from insoluble waste and concentrated by removing water under reduced pressure. YKS was mixed with powdered rodent chow (CE-2: CLEA Japan) at a concentration of 3% and fed to YKS-treated rats for 7 days prior to the test. This dose was chosen on the basis of effective doses of YKS in our previous study [33]. Previous studies have indicated that pre-administration of YKS may inhibit development of antinociceptive tolerance to morphine [33,34,43]. Thus, in this study, YKS administration was started 7 days before fentanyl injection. Rats that were not treated with YKS were fed powdered chow only.

Fentanyl (0.08 µg/kg) (Daiichisankyo, Tokyo, Japan) was injected intraperitoneally 10 min before pain induction using a 27 G hypodermic needle. This dose was determined by performing a preliminary experiment according to a previous study [38]. Rats not treated with fentanyl were intraperitoneally administered saline.

### 2.3. Assessment of Analgesia

The analgesic effects of fentanyl and YKS were examined using the formalin test and immunofluorescence staining of pERK. Subcutaneous injections of formalin have been widely used as an animal model of acute inflammatory pain [38,39,40].

#### 2.3.1. Formalin Test

Rats were randomly divided into five groups as follows: control (*n* = 7), formalin-injected (FOR; *n* = 7), YKS-treated + FOR (YKS; *n* = 9), fentanyl-treated + FOR (FEN; *n* = 9), and YKS + FEN + FOR (YKS + FEN; *n* = 9). The experimental protocol is shown in Table 2. Animals were housed individually in wire observation cages and habituated for 30 min. Ten minutes prior to the formalin test, animals were injected intraperitoneally with fentanyl or saline and returned to individual housing. Acute inflammatory pain was induced via an intraplantar injection of formalin (5%, 50 µL, Polysciences, Warrington, PA, US) into the right paw using a 30 G hypodermic needle. Rats in the control group were administered saline instead of formalin. Immediately after the injection, animals were returned to individual housing again, and the total time spent engaged in pain-related behavior was measured for the first 10 min (early phase), between 10 and 20 min (intermediate phase), and between 20 and 60 min (late phase) following the intraplantar injection of formalin or saline. Pain-related behavior was defined as paw shaking, licking, and lifting from the ground.

#### 2.3.2. Immunofluorescent Staining

The appearance of pERK in the dorsal horn was investigated using immunofluorescent staining. Rats were randomly divided into the same five groups (*n* = 4 in each group) and administered the same drugs as the formalin test. Forty-five minutes after formalin injection, rats were intraperitoneally anesthetized with pentobarbital sodium (50 μg/kg; Somnopentyl, Kyoritsu Seiyaku, Tokyo, Japan) and intracardially perfused with phosphate-buffered saline at pH 7.4 until all the blood had been removed from the system. After perfusion with 4% paraformaldehyde in 0.1 M phosphate-buffered saline, fifth lumbar spinal cord (L5) samples were harvested. Tissue specimens were immersed in 20% sucrose solution for 48 h and subsequently embedded in optimum cutting temperature compound (Tissue-Tek OCT, Sakura Finetek, Torrance, CA, USA), frozen, and cut into 15 µm sections using a cryostat (CM3050S, Leica Biosystems, Nussloch, Germany). Sections were incubated overnight at 4 °C with rabbit anti-pERK antibody (1:500, #4370, Cell Signaling Technology, Danvers, MA, USA). Sections were then incubated for 2 h with fluorophore-tagged secondary antibody (donkey anti-rabbit Alexa Fluor 555, 1:1000, #A31572, Thermo Fisher Scientific, Waltham, MA, USA). Nuclei were counterstained with DAPI (4′,6-diamidino-2-phenylindole, 1:1000, Thermo Fisher Scientific). Samples were imaged using a confocal laser scanning fluorescence microscope (FV1000D, Olympus, Tokyo, Japan), and cell co-localization of pERK and DAPI in the same area of laminae I–II were counted as pERK(+) cells by a third person who was not engaged in the staining process. The mean value was calculated using five sequential sections from each rat.

### 2.4. Cell Culture

HEK293T cells were cultured in Dulbecco’s modified Eagle’s medium (high glucose) with L-glutamine and phenol red (FUJIFILM Wako Pure Chemical, Osaka, Japan) containing 10% fetal bovine serum (#G121-6, JR Scientific, Woodland, CA, USA), penicillin/streptomycin (FUJIFILM Wako Pure Chemical), and GlutaMax (Gibco, Massachusetts, NY, USA) at 37 °C in humidified air containing 5% CO_2_. Cells were transfected with human TRPA1 cDNA (a generous gift from Dr. Yasuo Mori, Kyoto University) and 0.01 μg DsRed-express 2 vector (Takara Bio, Shiga, Japan) using Lipofectamine 3000 (Invitrogen, Waltham, CA, USA). Cells were replaced on cover slips after a 3 h incubation period and used 24–36 h after transfection.

### 2.5. Whole-Cell Patch-Clamp Recording

Transfected cells were identified by the red fluorescence signal excited by an LED illuminator, X-Cite XYLIS (Excelitas, Waltham, MA, USA). The bath solution contained 140 mM NaCl, 5 mM KCl, 2 mM CaCl_2_, 2 mM MgCl_2_, 10 mM glucose, and 10 mM HEPES (pH adjusted to 7.4 with NaOH). Pipette solution contained 140 mM CsCl, 5 mM 1.2-bis(o-aminophenoxy)ethane –N,N,N′,N′-tetraacetic acid (BAPTA), and 10 mM HEPES (pH adjusted to 7.4 with CsOH). Cells were treated with formalin (0.003%), FEN (10 μM), and YKS (1 mg/mL). YKS was decocted in standard bath solution at 60 °C for 15 min, and the supernatant after centrifugation (3000 rpm, 25 °C, 10 min) was used for the experiment. The TRPA1 currents were recorded in voltage-clamp mode using a Multiclamp 700B amplifier (Molecular Devices, California, USA), filtered at 1 kHz with a low-pass filter, and digitized with a Digidata 1550 B digitizer (Molecular Devices, San Jose, CA, USA). Data were acquired with pCLAMP 11 (Molecular Devices, California, USA). Pipette resistances were 3 ± 1 MΩ. The holding potential was −60 mV, and ramp pulses from −100 mV to +100 mV were applied for 300 ms every 5 s.

### 2.6. Statistical Analysis

Experimental data are presented as mean ± standard deviation. Statistical analyses were performed using one-way analysis of variance with Tukey’s test or the Tukey–Kramer post hoc test for comparisons (SPSS 24, IBM Japan, Tokyo, Japan). The *p*-value < 0.05 were considered statistically significant.

## 3. Results

### 3.1. Formalin Test

The analgesic effects of fentanyl and YKS were examined using the formalin test. The dose of fentanyl (0.08 µg/kg) was determined according to a previous study using the same experimental system. It is reported that 0.04 µg/kg of fentanyl has no effect on formalin-induced pain and 0.16 µg/kg significantly inhibits it [38]. We then performed a preliminary confirmation test using the doses of 0.04, 0.08, and 0.16 µg/kg, and the dose (0.08 µg/kg) that provided a moderate non-significant analgesic effect for the first 20 min was used because we would not be able to evaluate the effect of the drug combination if the dose (≥0.16 µg/kg) that provides significant analgesic effect was administered. With the formalin test, the effects are generally evaluated in two phases: phase I (0–10 min) and phase II (10–60 min). In the present study, we divided the total evaluation time into three phases: the early phase (0–10 min), the intermediate phase (10–20 min), and the late phase (20–60 min), because the effect of a single intraperitoneal administration of fentanyl lasts for approximately 30 min and administration was performed 10 min before pain induction.

In the early and intermediate phases, the duration of pain-related behavior was significantly increased according to the formalin injection; however, the increase was significantly inhibited in the YKS+FEN group (early phase, *p* < 0.01; intermediate phase, *p* < 0.05), but no significant effect was recognized in the YKS and FEN groups (Figure 2).

In the late phase, that is, after having lost the effects of fentanyl, the duration of pain-related behavior in the FEN group (1835.8 ± 415.4 s) was the longest and significantly increased compared with the YKS group (995.1 ± 220.3 s) (*p* < 0.01) (Figure 2).

### 3.2. Immunofluorescent Staining of pERK(+) Cells

The appearance of pERK in the late phase, 45 min after injection of formalin, was investigated to evaluate central sensitization. As a result, a similar tendency was observed with the formalin test in the late phase. Representative pictures are shown in Figure 3a. The number of pERK(+) cells in the FOR (5.96 ± 1.86 cells), FEN (6.83 ± 1.49 cells), and YKS+FEN (4.88 ± 1.42 cells) groups was significantly increased compared with the control group (0.83 ± 0.53 cells) (*p* < 0.01); however, the number of pERK(+) cells in the YKS group (3.19 ± 0.44 cells) was significantly lower (*p* < 0.05 vs. FOR, *p* < 0.01 vs. FEN) (Figure 3b).

### 3.3. Whole-Cell Patch-Clamp Recording of TRPA1 Currents

The TRPA1 channel is involved in formalin-induced pain sensation [39,40]. To investigate the pharmacological effects of fentanyl and YKS on TRPA1, we performed whole-cell patch-clamp recording in HEK293T cells expressing human TRPA1 and DsRed (Figure 4). TRPA1 was activated by 0.003% formalin, approximately the half maximal effective concentration according to a previous report [39]. First, we applied YKS and fentanyl, YKS, or fentanyl alone to check whether these medicines directly activated TRPA1. In these cells, exposure to YKS or fentanyl did not evoke a change in basal currents (Figure 4a,b) while the concomitant application of fentanyl and YKS slightly induced the current (Figure 4c). Subsequent addition of formalin activated TRPA1 and evoked a current with two phases, slow and rapid. However, the formalin-induced currents were observed in all experimental conditions. Thus, TRPA1 could be not inhibited by both fentanyl and YKS.

## 4. Discussion

Administration of fentanyl at a single high dose (e.g., ≥0.16 µg/kg; i.p. [38]) can inhibit pain induced by formalin; however, it may induce acute opioid tolerance and hyperalgesia [14,15,16,17,18,19,20,21,22]. We hypothesized that if the dose of fentanyl could be reduced, the risks may be mitigated. The present study examined the analgesic effect of a combination of fentanyl and YKS using a rat model of acute inflammatory pain. The effect of a single intraperitoneal administration of fentanyl lasted for approximately 20 min following formalin injection. In the early (0–10 min) and intermediate (1–20 min) phases, the combination of fentanyl and YKS significantly inhibited the time spent engaged in pain-related behavior, though the administration of fentanyl alone did not work effectively (Figure 2).

In the late phase (20–60 min); that is, after the effect of fentanyl had subsided, the duration of pain-related behavior in the YKS group was significantly decreased compared with the FEN group (*p* < 0.01). Moreover, that in the YKS group tended to be reduced compared with the FOR group (*p* = 0.061). However, when a strong analgesic effect is necessary, such as during the perioperative period, the use of opioid analgesics cannot be avoided. Although there was no significant difference (*p* = 0.057), that in the YKS+FEN group was shorter than the FEN group, and therefore combined use of YKS and fentanyl is thought to be clinically useful.

To elucidate its mechanism of action, we investigated the expression of pERK in the dorsal horn of the spinal cord. Moreover, we investigated the influence of YKS and fentanyl on TRPA1 in vitro. TRPA1, a calcium-permeable non-selective cation channel, is activated by various chemicals, including irritant exogenous ligands and endogenous ligands produced by inflammation [44]. TRPA1 expression in the nociceptive primary sensory nerve is related to the reception of noxious stimuli and signal transduction to the secondary sensory nerve, and activation of TRPA1 induces hyperalgesia [44]. Pain in phase I (0–10 min) of the formalin test is mediated by the activation of TRPA1, and is attenuated by TRPA1-selective antagonists [39,40]. Therefore, the influence of YKS and fentanyl on TRPA1 was investigated. According to the patch-clamp recording, both YKS and fentanyl had no antagonistic activity on TRPA1 (Figure 4). One report suggested that morphine activates TRPA1 [45], and influences of opioids on TRPA1 may be different depending on the type of opioid. In this study, we could not reveal the mechanism of analgesic efficacy of the combination of fentanyl and YKS in the early (0–10 min) and intermediate (10–20 min) phases. We previously reported that the administration of YKS increased the secretion of oxytocin in rats with acute psychological stress [46], and Gamal-Eltrabily et al. [47] reported the injection of oxytocin inhibited formalin-induced pain. Including this, further studies concerning the action mechanism are needed.

Mitogen-activated protein kinase (MAPK) pathways play an important role in nociceptive and neuropathic pain [42,48]. In phase II of the biphasic pain response caused by formalin, the ERK pathway was activated in the central nucleus, which may be involved in nociceptive plasticity [49]. U0126, a specific inhibitor of the ERK pathway, suppressed persistent pain induced by formalin [50]. Moreover, pERK(+) neurons in the spinal cord were increased following remifentanil infusion [51]. Thus, we explored the influence on the ERK pathway as the mechanism underlying the therapeutic effect of YKS. Our results show that YKS suppressed pERK expression, which may be related to the analgesic effect of YKS in the late phase. The anti-inflammatory action of saikosaponin A isolated from *Bupleuri* radix [52], the anti-inflammatory and anti-tumorigenic effects of total flavonoids from *Glycyrrhizae* radix [53], and the neuroprotective effects of liquiritigenin isolated from *Glycyrrhizae* radix [54] are exerted by inhibition of the ERK pathway. In the future, we will examine whether these components inhibit the ERK pathway in the present experimental system.

OIH, including RIH and FIH, is thought to be a complex physiological response involving glial cell activity [6,13], neuroinflammation [55], opioid receptor desensitization [12], and NMDA receptor activation [9,10,11,18]. YKS has an ameliorative effect on glutamate clearance in astrocytes and an antagonistic action at the NMDA receptor [35,36,37]. Additionally, we previously reported that administration of YKS attenuated the development of antinociceptive morphine tolerance, and that suppression of glial cell activation in the spinal cord and mesencephalon may be one mechanism underlying this phenomenon [33,34]. These mechanisms are also thought to contribute to the preventative effect of YKS on the development of FIH. In this study, obvious FIH was not observed in the late phase, possibly because the dose of fentanyl was low; thus, further studies should be conducted using higher doses of fentanyl.

With respect to the analgesic effect of YKS, almost all clinical and basic studies investigated chronic pain [25,26,27,28,29,30,31,32], and no studies have investigated acute pain. The results from this study suggest that combination use with opioid analgesics might contribute to a reduction in opioid dose and prevention of paradoxical reactions following opioid use; however, YKS alone cannot be expected to provide a sufficient analgesic effect against acute inflammatory pain.

## 5. Conclusions

Fentanyl may induce acute opioid tolerance and postoperative hyperalgesia when administered at a single high dose. In this study, although fentanyl, the dose of which (0.08 µg/kg) cannot provide a significant analgesic effect, was used, combined use of YKS and fentanyl could significantly inhibit pain in the early and intermediate phases of the formalin test. Our findings suggest that YKS could reduce the use of fentanyl and the combined use considered clinically useful.

## Figures and Tables

**Figure 1 medicines-07-00075-f001:**
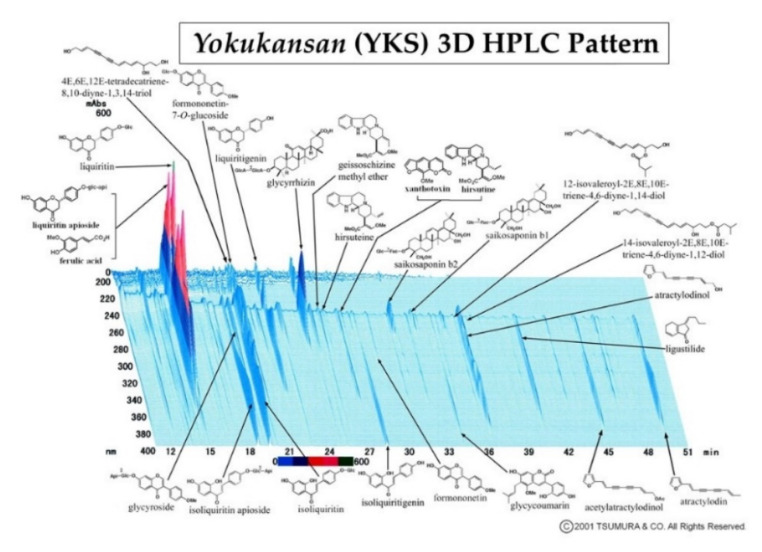
Three-dimensional high-performance liquid chromatography (3D-HPLC) profile chart of the major chemical compounds in *Yokukansan* (YKS).

**Figure 2 medicines-07-00075-f002:**
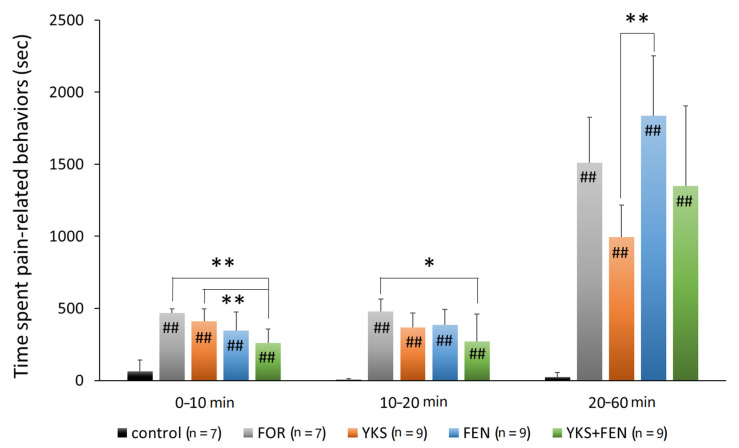
Duration of pain-related behavior with the formalin test. The combination of *Yokukansan* (YKS) and fentanyl (FEN) significantly inhibited pain-related behavior during the early phase (0–10 min after formalin injection) and the intermediate phase (10–20 min). The duration in the YKS group was significantly shorter compared with the FEN group during the late phase. FOR, formalin-injected group; YKS, YKS-treated+FOR group; FEN, fentanyl-treated+FOR group; YKS+FEN, YKS+FEN+FOR group. Mean ± SD. ^##^
*p* < 0.01 (vs. control), * *p* < 0.05, ** *p* < 0.01 (Tukey–Kramer test).

**Figure 3 medicines-07-00075-f003:**
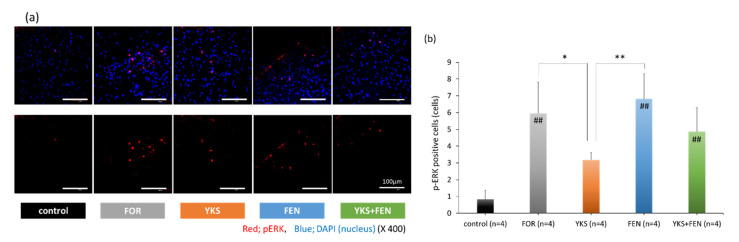
Immunofluorescent staining of pERK(+) cells. (**a**) Appearance of pERK in the dorsal horn. The upper row includes pERK and nuclei, and the lower row includes only pERK. Red, pERK; blue, DAPI (nuclei). (**b**) The number of pERK(+) cells in the FOR, FEN, and YKS+FEN groups was significantly increased compared with the control group (*p* < 0.01); however, the number of pERK(+) cells in the YKS group was significantly lower (*p* < 0.05 vs. FOR, *p* < 0.01 vs. FEN). FOR, formalin-injected group; YKS, YKS-treated+FOR group; FEN, fentanyl-treated+FOR group; YKS+FEN, YKS+FEN+FOR group. Mean ± SD. ^##^
*p* < 0.01 (vs. control), * *p* < 0.05, ** *p* < 0.01 (Tukey’s test).

**Figure 4 medicines-07-00075-f004:**
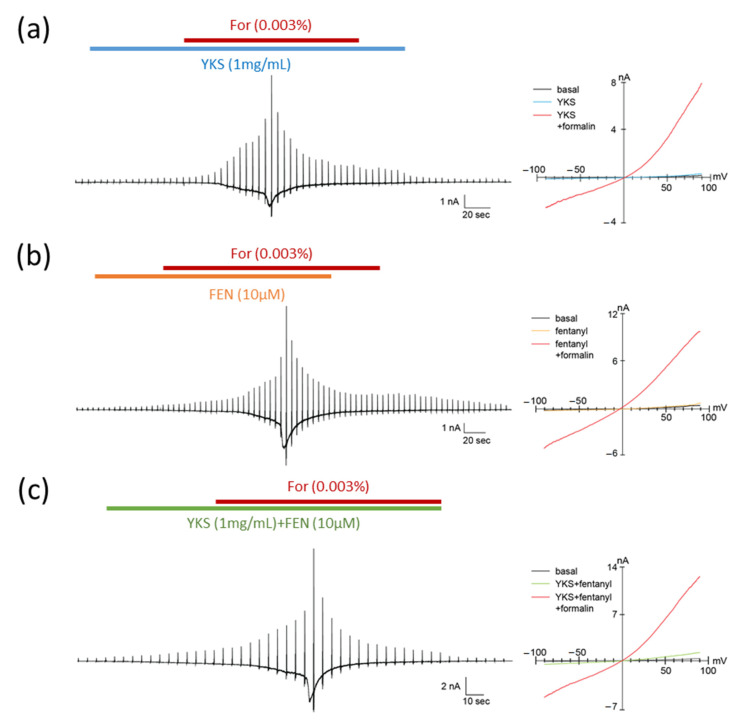
Pharmacological effects of formalin, fentanyl, and *Yokukansan* on TRPA1. Typical traces (left) and current–voltage relationships (right) of formalin-induced TRPA1 currents in HEK293T cells expressing human TRPA1. The concentrations of formalin (For), fentanyl (FEN), and *Yokukansan* (YKS) were 0.003%, 10 μM, and 1 mg/mL, respectively. YKS (**a**) or FEN (**b**), and YKS and FEN (**c**) were pretreated for 1 min before the concomitant administration of formalin. TRPA1 was not inhibited by both fentanyl and YKS. The holding potential was −60 mV, and ramp pulses (−100 to +100 mV, 300 ms) were applied every 5 s.

**Table 1 medicines-07-00075-t001:** The component galenicals of *Yokukansan* (YKS).

*Uncariae cum Uncis* ramulus	3.0 g
*Cnidii* rhizoma	3.0 g
*Bupleuri* radix	2.0 g
*Atratylodis Lanceae* rhizoma	4.0 g
*Poria*	4.0 g
*Angelicae* radix	3.0 g
*Glycyrrhizae* radix	1.5 g

Weights indicate relative amounts mixed.

**Table 2 medicines-07-00075-t002:** The experimental protocol.

Groups	Days 1–7	Day 8		
		10 min before Test	Formalin Test	
control	Powdered chow	Saline (i.p.)	Saline (50 µL; s.c.)	Observation of pain-related behavior (60 min)
FOR	Powdered chow	Saline (i.p.)	Formalin (5%, 50 µL; s.c.)	
YKS	Powdered chow mixed with YKS (3%)	Saline (i.p.)	Formalin (5%, 50 µL; s.c.)	
FEN	Powdered chow	Fentanyl (0.08 µg/kg; i.p.)	Formalin (5%, 50 µL; s.c.)	
YKS+FEN	Powdered chow mixed with YKS (3%)	Fentanyl (0.08 µg/kg; i.p.)	Formalin (5%, 50 µL; s.c.)	

Groups are as follows: control, formalin-injected (FOR), YKS-treated + FOR (YKS), fentanyl-treated + FOR (FEN), and YKS + FEN + FOR (YKS + FEN). YKS was mixed with powdered rodent chow at a concentration of 3% and fed to YKS-treated rats for 7 days prior to the test. YKS, *Yokukansan*; i.p., intraperitoneal injection; s.c., subcutaneous injection.

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
