# Peer review of "Analgesic Efficacy of a Combination of Fentanyl and a Japanese Herbal Medicine “*Yokukansan*” in Rats with Acute Inflammatory Pain"

_medicines, 2020, doi:10.3390/medicines7120075_

Round 1

Reviewer 1 Report

ABSTRACT: Line 19, the first sentence - Give an explanation as you did in the Introductory section (lines 40-42).

Line 26 - Correct intermediate phase (10-20 min)

Line 31 - Correct the sentence - "... and the increase in pain in the late phase was inhibited compared with the FEN+FOR group"

The last correction should be put into CONCLUSION section too,  Line 306.

In Ca2+ imaging experiments, for activation HEK293T cells, should be add new data by formalin with corresponding control, fentanyl and Yokukansan trials.  

Author Response

We would like to thank you for your review.
At first, we have to apologize. We thank to reviewer’s constructiveness comments. According to the comments of you and reviewer 2, we checked formalin-induced TRPA1 activation by calcium imaging; however, the results suggested that formalin nonspecifically increased the intracellular calcium concentration in HEK293T cells. And then, according to the comment of reviewer 3, we checked fentanyl and YKS effects on TRPA1 current induced by formalin in patch-clamp recording. The results of patch-clamp recording suggested that both fentanyl and YKS did not inhibit TRPA1. We think that the patch-clamp recording is more suitable method to pharmacologically investigate ion channel function. In our case, the data of calcium imaging could contain unknown artifact while we cannot conclude the details. Therefore, we decided to remove the data of calcium imaging to the results of TRPA1 current. Thus, we majorly modified the manuscript.

We have revised my manuscript as follows.

Point 1: Line 19, the first sentence - Give an explanation as you did in the Introductory section (lines 40-42).

Response 1: We added the explanation as you as you have pointed out.

Point 2: Line 26 - Correct intermediate phase (10-20 min)
Response 2: (Line 28) We corrected. Thank you so much.

Point 3: Line 31 - Correct the sentence - "... and the increase in pain in the late phase was inhibited compared with the FEN+FOR group" The last correction should be put into CONCLUSION section too, Line 306.

Response 3: (Line 33) Our interpretation of the results was taken exception to by another reviewer. Then, we modified the conclusions.

Point 4: In Ca2+ imaging experiments, for activation HEK293T cells, should be add new data by formalin with corresponding control, fentanyl and Yokukansan trials. 

Response 4: As mentioned above, we checked fentanyl and YKS effects on TRPA1 current induced by formalin in patch-clamp recording. And, the results suggested that both fentanyl and YKS did not directly inhibit TRPA1. Therefore, we majorly modified the manuscript.

Reviewer 2 Report

The authors investigate in their manuscript the hypothesis that a Japanese herbal medicine “Yokukansan” (YKS) might inhibit the development of fentanyl-induced hyperalgesia (FIH). Fentanyl is widely used in general anesthesia but its administration is limited due to the FIH development. Overcoming this side effect would certainly be valuable for clinical practice. The authors showed that administration of YKS together with fentanyl shorten the time spent engaged in pain-related behavior in the early and intermediate phases of formalin test. These results suggest that combination of fentanyl and YKS might be the feasible approach how to reduce the necessary dose of fentanyl and as consequence also the fentanyl side effects.

The authors clearly describe the links of evidence that lead them to their hypothesis. This study follows up a couple other studies from the same laboratory that also concern the effects of YKS in the pain management.

In the present study the authors used three experimental approaches. I don't feel qualified to judge about the part that concerns immunofluorescent staining.

The pain-related behavior test seems to be correctly done with appropriate number of animals and described with sufficient details. However, I have several major comments concerning the interpretation of the obtained data.

The major problem with the result of formalin test is that, in fact, it did not manifest the FIH. Probably due to chosen dose of fentanyl. The FOR and FEN groups in the late phase are not significantly different (Figure 2) which means that the FIH is not observed and YKS effect on FIH cannot be evaluated. Due to this, please revisit the formulation of Conclusion section in the abstract on line 31: “the increase in pain in the late phase was inhibited.” There is no increase that can be inhibited.

Likewise the sentence (occurring twice at line 193 and 200) needs to be revisited: “Although there were no significant differences, the duration of pain-related behavior in the YKS group was shorter compared with the FOR group (1509.1 ± 315.5 sec) (p = 0.061), and the duration in the YKS+FEN group (1351.1 ± 552.9 sec) was shorter compared with the FEN group (p = 0.057) (Figure 2).”

In addition, the first part of the sentence (Line 30) states: “In combination with YKS, the analgesic effect of fentanyl was enhanced in the early and intermediate phases.” This statement is not supported by the data in Figure 2 because it refers to FEN and YKS+FEN groups, which are not significantly different. Moreover, at the presented dose, the fentanyl itself has no analgesic effect (no difference between FOR and FEN groups).

The dose of fentanyl used is “The dose that provided a moderate non-significant analgesic effect” (Line 247). The reasons why such dose has been selected should be explained in more details and moved to the results section.

Line 255: “FEN group was increased compared with the other groups.” This needs to be formulated more carefully. FEN was increased compared only with YKS group.

Line 27-28: “In the late phase (20–60 min), pain-related behavior in the FEN group was increased compared with other groups but was alleviated by YKS.” This is the same case as in the previous comment.

Line 258 – 261:  “When considering FIH, although there was no significant difference between the FEN and YKS+FEN groups (p = 0.057), YKS tended to ameliorate the development of FIH, and combined use of YKS and fentanyl is thought to be clinically useful.” As I stated above the data from the late phase do not capture the FIH phenomenon and cannot be interpreted this way.

The third experimental approach used in this study was the calcium imaging experiment. The results presented in the Figure 4 are insufficiently presented and incorrectly interpreted.

The authors should state in the Materials and Methods section how many individual transfection were used for each experimental group. The authors should also explain how they identified the positively transfected cells and describe the system for the solution application.  

Line 226: “Results are expressed as the relative ratio of the increase over baseline, normalized to the maximum effect induced by ionomycin (5 μM) added at the end of each experiment (Figure 4).” The authors should include the representative traces for each experimental group into figure 4 to clarify the duration of the AITC application and subsequent washout.

Line 233: “In YKS-treated cells, AITC-induced responses were also significantly suppressed (n = 30; 0.56 ± 0.06, p < 0.05) (Figure 4b).” It should be clearly stated that the difference occurs after the YKS washout.

Line 271: “According to the calcium imaging assay, both YKS and fentanyl had antagonistic activity on TRPA1 (Figure 4).” This statement is not supported by the data. As the co-application of AITC and fentanyl or YKS evoked the same response as AITC alone. The significant difference after washout of fentanyl of YKS cannot be interpreted as antagonistic effect on an ion channel.

I have some concerns about the chosen AITC concentration. The lower AITC concentration or the usage of some less potent TRPA1 agonist may have more chance to capture the effect of fentanyl and YKS on TRPA1 activation.

Line 303-307: The conclusion section needs to be revisited to address the above comments.

Minor comments:

Line 255- 256 : “This suggested that FIH.” The sentence seems to be unfinished.

Line 150: “co-expressing pERK and DAPI” DAPI as a fluorescent stain cannot be expressed.  The authors probably meant co-localization?

There are no y-axis ticks on the graphs in Figure 2 and Figure 3.

Author Response

I would like to thank you for your detailed review. I have revised my manuscript as follows.

  The authors investigate in their manuscript the hypothesis that a Japanese herbal medicine “Yokukansan” (YKS) might inhibit the development of fentanyl-induced hyperalgesia (FIH). Fentanyl is widely used in general anesthesia but its administration is limited due to the FIH development. Overcoming this side effect would certainly be valuable for clinical practice. The authors showed that administration of YKS together with fentanyl shorten the time spent engaged in pain-related behavior in the early and intermediate phases of formalin test. These results suggest that combination of fentanyl and YKS might be the feasible approach how to reduce the necessary dose of fentanyl and as consequence also the fentanyl side effects.
  The authors clearly describe the links of evidence that lead them to their hypothesis. This study follows up a couple other studies from the same laboratory that also concern the effects of YKS in the pain management.
  In the present study the authors used three experimental approaches. I don't feel qualified to judge about the part that concerns immunofluorescent staining.
  The pain-related behavior test seems to be correctly done with appropriate number of animals and described with sufficient details. However, I have several major comments concerning the interpretation of the obtained data.

Point 1: The major problem with the result of formalin test is that, in fact, it did not manifest the FIH. Probably due to chosen dose of fentanyl. The FOR and FEN groups in the late phase are not significantly different (Figure 2) which means that the FIH is not observed and YKS effect on FIH cannot be evaluated. Due to this, please revisit the formulation of Conclusion section in the abstract on line 31: “the increase in pain in the late phase was inhibited.” There is no increase that can be inhibited.
Point 2: Likewise the sentence (occurring twice at line 193 and 200) needs to be revisited: “Although there were no significant differences, the duration of pain-related behavior in the YKS group was shorter compared with the FOR group (1509.1 ± 315.5 sec) (p = 0.061), and the duration in the YKS+FEN group (1351.1 ± 552.9 sec) was shorter compared with the FEN group (p = 0.057) (Figure 2).”
Point 3: In addition, the first part of the sentence (Line 30) states: “In combination with YKS, the analgesic effect of fentanyl was enhanced in the early and intermediate phases.” This statement is not supported by the data in Figure 2 because it refers to FEN and YKS+FEN groups, which are not significantly different. Moreover, at the presented dose, the fentanyl itself has no analgesic effect (no difference between FOR and FEN groups).
Point 4: Line 255: “FEN group was increased compared with the other groups.” This needs to be formulated more carefully. FEN was increased compared only with YKS group.
Point 5: Line 27-28: “In the late phase (20–60 min), pain-related behavior in the FEN group was increased compared with other groups but was alleviated by YKS.” This is the same case as in the previous comment.
Point 6: Line 258 – 261:  “When considering FIH, although there was no significant difference between the FEN and YKS+FEN groups (p = 0.057), YKS tended to ameliorate the development of FIH, and combined use of YKS and fentanyl is thought to be clinically useful.” As I stated above the data from the late phase do not capture the FIH phenomenon and cannot be interpreted this way.

We thank to your precise comments Point 1~6. As you have pointed out, we mistook the interpretation of the results.

Response 1: (Line 34) This sentence was deleted.

Response 2: (Line 219) This was also deleted.

Response 3: (Line 33) This was also deleted.

Response 4: (Line 298) This was also deleted.

Response 5: (Line 28) This sentence was corrected.

Response 6: (Line 303) This sentence was corrected.  

Point 7: The dose of fentanyl used is “The dose that provided a moderate non-significant analgesic effect” (Line 247). The reasons why such dose has been selected should be explained in more details and moved to the results section.

Response 7: We added the sentence and moved to the results section (Line 202).

We performed a preliminary confirmation test using the doses of 0.04, 0.08 and 0.16 µg/kg (n=3 each), and 0.16 µg/kg significantly inhibited formalin-induced pain for the first 20 min. We then used the dose (0.08 µg/kg) that could not provide a significant analgesic effect, because one purpose of the study was to evaluate the effect of the drug combination, we thought we would not be able to evaluate the effect if the dose (≥0.16 µg/kg) that provides an significant analgesic effect was administered.

Point 8: The third experimental approach used in this study was the calcium imaging experiment. The results presented in the Figure 4 are insufficiently presented and incorrectly interpreted.
The authors should state in the Materials and Methods section how many individual transfection were used for each experimental group.

Response 8: As described below, we majorly modified our manuscript due to additional experiment. 

Point 9: The authors should also explain how they identified the positively transfected cells and describe the system for the solution application.

Response 9: (Line 165) We added sentences in methods “ 2.4. Cell Culture” according to your comment.

Point 10: Line 226: “Results are expressed as the relative ratio of the increase over baseline, normalized to the maximum effect induced by ionomycin (5 μM) added at the end of each experiment (Figure 4).” The authors should include the representative traces for each experimental group into figure 4 to clarify the duration of the AITC application and subsequent washout.

Response 10: In our case, we took each picture at the described time points because the time course of TRPA1 activity has been known very well by many previous reports, rapid rising of intracellular calcium concentration and unclear reduction after washing out. Anyway, we removed this figure and results in revision.

Point 11: Line 233: “In YKS-treated cells, AITC-induced responses were also significantly suppressed (n = 30; 0.56 ± 0.06, p < 0.05) (Figure 4b).” It should be clearly stated that the difference occurs after the YKS washout.
Line 271: “According to the calcium imaging assay, both YKS and fentanyl had antagonistic activity on TRPA1 (Figure 4).” This statement is not supported by the data. As the co-application of AITC and fentanyl or YKS evoked the same response as AITC alone. The significant difference after washout of fentanyl of YKS cannot be interpreted as antagonistic effect on an ion channel.
I have some concerns about the chosen AITC concentration. The lower AITC concentration or the usage of some less potent TRPA1 agonist may have more chance to capture the effect of fentanyl and YKS on TRPA1 activation.

Response 11: (Line 168) We thank to your constructiveness comments. We checked formalin-induced TRPA1 activation by calcium imaging, however the results suggested that formalin nonspecifically increased the intracellular calcium concentration in HEK293T cells. According to comments of you and other reviewers, we checked fentanyl and YKS effects on TRPA1 current induced by formalin in patch-clamp recording. In this experiment, we used formalin at approximately EC50 according to previous report (McNamara et al, PNAS, 2007, PMID:17686976). The results of patch-clamp recording suggested that both fentanyl and YKS did not inhibit TRPA1. We think that the patch-clamp recording is more suitable method to pharmacologically investigate ion channel function. In our case, the data of calcium imaging could contain unknown artifact while we cannot conclude the details. Therefore, we decided to remove the data of calcium imaging to the results of TRPA1 current. Thus, we majorly modified the manuscript.

Point 12: Line 303-307: The conclusion section needs to be revisited to address the above comments.
Response 12: (Line 361) We revised it. Thank you again.

Minor comments:

Point 13: Line 255- 256: “This suggested that FIH.” The sentence seems to be unfinished.
Response 13: (Line 300) As you mentioned above, we modified the interpretation of the results.

Point 14: Line 150: “co-expressing pERK and DAPI” DAPI as a fluorescent stain cannot be expressed.  The authors probably meant co-localization?
Response 14: (Line 156) Just as you said it. we corrected it.

Point 15: There are no y-axis ticks on the graphs in Figure 2 and Figure 3.

Response 15: We changed them.

Reviewer 3 Report

Fentanyl and remifentanil are potent, short-lasting µ-opioid receptor agonists with limited use due to their proclivity to induce acute opioid tolerance and post-operative hyperalgesia. Previous studies have linked glutamate release, NMDA receptor activation, and microglia/astrocyte induction to remifentanil-induced hyperalgesia (RIH). Yokukansan (YKS) is a Japanese traditional herbal comprised of seven herbs, and recently it has been reported that YKS is effective in managing pain disorders. YKS has been shown to impact glutamate clearance in astrocytes and act as an antagonist at the NMDA receptor, making it a possibility that YKS may inhibit the development of fentanyl-induced hyperalgesia (FIH). The inflammatory pain model induced by formalin injection into the paw was used to ascertain the combined treatment of YKS and fentanyl. Pain-related behavior was measured as paw lifting, licking, and shaking. After treatment, the expression of phosphorylated extracellular signal-regulated kinase (pERK) in the spinal dorsal horn was analyzed, as pERK has been shown to be an important molecule in pain signaling. Interestingly, the YKS+FEN group experienced shorter periods of pain-related behavior compared to the other groups during both the early and intermediate phase. As predicted, the FEN group experienced heightened lengths of pain-related behavior during the late phase, likely indicating hyperalgesia. YKS+FEN decreased the time spent, albeit non-significantly. YKS alone also decreased the time spent during the late period compared to formalin, which would imply that YKS alone does not induce hyperalgesia. Figure 3 showed that the number of cells that were positive for pERK was significantly reduced compared to both the FOR group and FEN group, indicating that YKS could reduce pERK expression. Figure 4 showed that application of AITC, a TRPA1 agonist, after application of either fentanyl or YKS elicited smaller responses, implicating an alteration in TRPA1 signaling.

Major Issues

  • It’s unclear why the authors chose to not use formalin as the compound to activate TRPA1, or at least provide it as a second compound for Figure 4. Since the story is about alterations in the formalin-induced pain-related behaviors, and since formalin activates TRPA1 similarly to AITC, can the authors provide rationale for not conducting the experiment with formalin?
  • It’s also unclear why for Figure 4 the combination of FEN + YKS was not tested, as the rest of the manuscript is discussing the improved analgesic efficacy in all three stages of the formalin test. The authors should conduct the same experiment in HEK293T cells with co-application of YKS and FEN to determine whether there is a further decrease in signaling through TRPA1.
  • The author’s main question is whether the dose of fentanyl can be decreased with the co-administration of YKS; however, the dose of fentanyl was never adjusted, and this question was never examined specifically. The authors should conduct an experiment where they adjust the dose of fentanyl with the combination of YKS and determine if the same analgesic efficacy can be attained while preventing hyperalgesia in the late period before making this question the central question of the manuscript.

Minor Issues

  • In the introduction, it states: “YKS is approved by the Japanese Ministry of Health, Labor, and Welfare.” What exactly is it approved for?
  • In the methods for calcium imaging, it says Fura-8 AM was used to measure intracellular calcium levels, and then subsequently says that Fluo-8 was measured in normal bath solution. Figure 4 is also measuring Fluo-8 fluorescence – where is Fura-8 AM being used?
  • Would application of AITC, another activator of TRPA1, cause a similar increase in pERK?
  • It’s interesting that both fentanyl and YKS (probably through liquiritin) inhibited TRPA1 activity. I would be interested in seeing if this effect was similar in patch-clamp studies. Further research would need to be conducted to determine whether this effect is due to direct binding to TRPA1 or through another effector, such as G-proteins.
  • It seems unlikely that inhibition of TRPA1 is impacting pERK levels. If this were the case, we would expect similar pERK levels in FEN treated cells compared to YKS treated cells. Unless fentanyl’s hyperalgesia effect is separate from TRPA1.

Author Response

I would like to thank you for your review. I have revised my manuscript as follows.

Fentanyl and remifentanil are potent, short-lasting µ-opioid receptor agonists with limited use due to their proclivity to induce acute opioid tolerance and post-operative hyperalgesia. Previous studies have linked glutamate release, NMDA receptor activation, and microglia/astrocyte induction to remifentanil-induced hyperalgesia (RIH). Yokukansan (YKS) is a Japanese traditional herbal comprised of seven herbs, and recently it has been reported that YKS is effective in managing pain disorders. YKS has been shown to impact glutamate clearance in astrocytes and act as an antagonist at the NMDA receptor, making it a possibility that YKS may inhibit the development of fentanyl-induced hyperalgesia (FIH). The inflammatory pain model induced by formalin injection into the paw was used to ascertain the combined treatment of YKS and fentanyl. Pain-related behavior was measured as paw lifting, licking, and shaking. After treatment, the expression of phosphorylated extracellular signal-regulated kinase (pERK) in the spinal dorsal horn was analyzed, as pERK has been shown to be an important molecule in pain signaling. Interestingly, the YKS+FEN group experienced shorter periods of pain-related behavior compared to the other groups during both the early and intermediate phase. As predicted, the FEN group experienced heightened lengths of pain-related behavior during the late phase, likely indicating hyperalgesia. YKS+FEN decreased the time spent, albeit non-significantly. YKS alone also decreased the time spent during the late period compared to formalin, which would imply that YKS alone does not induce hyperalgesia. Figure 3 showed that the number of cells that were positive for pERK was significantly reduced compared to both the FOR group and FEN group, indicating that YKS could reduce pERK expression. Figure 4 showed that application of AITC, a TRPA1 agonist, after application of either fentanyl or YKS elicited smaller responses, implicating an alteration in TRPA1 signaling.

Major Issues

Point 1: It’s unclear why the authors chose to not use formalin as the compound to activate TRPA1, or at least provide it as a second compound for Figure 4. Since the story is about alterations in the formalin-induced pain-related behaviors, and since formalin activates TRPA1 similarly to AITC, can the authors provide rationale for not conducting the experiment with formalin?

It’s also unclear why for Figure 4 the combination of FEN + YKS was not tested, as the rest of the manuscript is discussing the improved analgesic efficacy in all three stages of the formalin test. The authors should conduct the same experiment in HEK293T cells with co-application of YKS and FEN to determine whether there is a further decrease in signaling through TRPA1.

Response 1: AITC is a major agonist to experimentally activate TRPA1 in vitro. Furthermore, formalin nonspecifically increased the intracellular calcium concentration in HEK293T cells. Therefore, we used AITC in calcium imaging. However, we agree the reviewer's comment emphasizing the in vivo experiments. We checked fentanyl and YKS effects on TRPA1 current induced by formalin in patch-clamp recording. And, the results suggested that both fentanyl and YKS did not directly inhibit TRPA1. Therefore, we are terribly sorry, but we majorly modified the manuscript.

Point 2: The author’s main question is whether the dose of fentanyl can be decreased with the co-administration of YKS; however, the dose of fentanyl was never adjusted, and this question was never examined specifically. The authors should conduct an experiment where they adjust the dose of fentanyl with the combination of YKS and determine if the same analgesic efficacy can be attained while preventing hyperalgesia in the late period before making this question the central question of the manuscript.
Response 2: (Line 201) According to a previous study [38], we performed a preliminary confirmation test using the doses of 0.04, 0.08 and 0.16 µg/kg (n=3 each), and the dose (0.08 µg/kg) that provided a moderate non-significant analgesic effect for the first 20 min was used. However, as you said, we ought to have conducted and compared using some dose. We will investigate in the future.

Minor Issues

Point 3: In the introduction, it states: “YKS is approved by the Japanese Ministry of Health, Labor, and Welfare.” What exactly is it approved for?

Response 3: (Line 62) In Japan, YKS is not folk medicine and is officially approved as an ethical pharmaceutical. I added it in the introduction.

Point 4: In the methods for calcium imaging, it says Fura-8 AM was used to measure intracellular calcium levels, and then subsequently says that Fluo-8 was measured in normal bath solution. Figure 4 is also measuring Fluo-8 fluorescence – where is Fura-8 AM being used?

Response 4: Fluo-8 AM is a chemical compound composed by Fluo-8 and acetoxymethyl (AM). This compound is plasma membrane permeable, however the binding between Fluo-8 and AM is hydrolyzed inside of cells by esterase. Fluo-8 cannot permeate membrane. Therefore, we can measure the Fluo-8 fluorescent changes by intracellular calcium concentration.

Point 5: Would application of AITC, another activator of TRPA1, cause a similar increase in pERK?

Response 5: It is reported that TRPA1 may be involved in an activation of ERK pathway.

Griggs RB, Laird DE, Donahue RR, Fu W, Taylor BK. Methylglyoxal Requires AC1 and TRPA1 to Produce Pain and Spinal Neuron Activation. Front Neurosci. 2017;11:679.

But, the results of in vitro experiment was modified, we did not add the comment about this.

Point 6: It’s interesting that both fentanyl and YKS (probably through liquiritin) inhibited TRPA1 activity. I would be interested in seeing if this effect was similar in patch-clamp studies. Further research would need to be conducted to determine whether this effect is due to direct binding to TRPA1 or through another effector, such as G-proteins.

It seems unlikely that inhibition of TRPA1 is impacting pERK levels. If this were the case, we would expect similar pERK levels in FEN treated cells compared to YKS treated cells. Unless fentanyl’s hyperalgesia effect is separate from TRPA1.

Response 6: As describe above, we added the results by patch-clamp recording, and majorly modified our manuscript. Thank you again.

Round 2

Reviewer 1 Report

Now, after revisions should be accepted for publishing. 

Author Response

I would like to thank you for your review.

Now, after revisions should be accepted for publishing.

Response: Thank you so much.

Reviewer 2 Report

The authors incorporated the necessary changes into their manuscript and substantially improved the interpretation of the obtained data. For obvious reasons, the additional path-clamp experiments seems to be done in a hurry. Control measurements with formalin alone and statistical evaluation of the data presented in figure 4 are missing. However, presented experiments appear to be sufficient to illustrate that FEN and YKS have no apparent effect on formalin-induced TRPA1 currents.

Minor comment:

Line 241: The dose of YKS is stated 0.5 mg/ml but in the Figure 4a and 4c is stated 1 mg/ml.

The figure 3 in my pdf version is split.

Please check the scale bars in Figure 4. Because the red IV line in the right panel goes to 8 nA, but the biggest positive current in left panel is 0.8 nA. What is correct?

Author Response

I would like to thank you for your review.

The authors incorporated the necessary changes into their manuscript and substantially improved the interpretation of the obtained data. For obvious reasons, the additional path-clamp experiments seems to be done in a hurry. Control measurements with formalin alone and statistical evaluation of the data presented in figure 4 are missing. However, presented experiments appear to be sufficient to illustrate that FEN and YKS have no apparent effect on formalin-induced TRPA1 currents.

Minor comment:

Point 1: Line 241: The dose of YKS is stated 0.5 mg/ml but in the Figure 4a and 4c is stated 1 mg/ml.

Response 1: 1 mg/ml was correct. We corrected.

Point 2: The figure 3 in my pdf version is split.

Response 2: We revised the figure.

Point 3: Please check the scale bars in Figure 4. Because the red IV line in the right panel goes to 8 nA, but the biggest positive current in left panel is 0.8 nA. What is correct?

Response 3: We revised the scale bars. Thank you so much.

Reviewer 3 Report

Thank you for addressing our concerns.

Author Response

I would like to thank you for your review.

Thank you for addressing our concerns.

Response: Thank you so much.